# Quantification of the radiative forcing of contrails embedded in cirrus clouds

Torsten Seelig [1] ✉, Kevin Wolf [1], Nicolas Bellouin [2] & Matthias Tesche [1]

Aviation leads to the emission of $CO_2$ but also exerts non-$CO_2$ effects on climate, such as line-shaped condensation trails (contrails) and contrail cirrus that are known to cause warming. However, little is known about the climate effect of contrails that form in already existing cirrus clouds, where conditions for contrail formation are found most often. Here, we infer the local net radiative forcing of around 40,000 embedded contrails by combining aircraft position data with height-resolved cloud observations from spaceborne lidar. Considering the period from 2015 to 2021, we find an annual mean local warming effect of 60 mW m$^{-2}$. Expanding these findings to the global scale suggests an annual global mean net radiative forcing of embedded contrails on the order of 5 mW m$^{-2}$. This corresponds to around 10% of the current estimate of the climate impact of line-shaped contrails and suggests that embedded contrails are a non-negligible contributor to aviation's impact on climate.

Aviation affects the Earth's energy balance in multiple ways[1]. In addition to carbon dioxide ($CO_2$) emissions, the climate impact of aviation is also connected to a number of non-$CO_2$ effects[2]. Best known is the formation of optically thin aircraft condensation trails (contrails) and contrail cirrus, which, on global average, exert a positive global net radiative forcing (RF) on the order of 50 mW m$^{-2}$, i.e., have a warming effect[2–6].

A contrail has to persist for at least 10 min to be defined as *Cirrus homogenitus*. Longer-lasting (persistent) contrails may spread into contrail cirrus, which is defined as *Cirrus homomutatus*[7]. The properties, development, and RF of line-shaped contrails and contrail cirrus have been studied for a long time in both satellite observations and modeling e.g.,[8–15]. While there is growing evidence that conditions for contrail formation often coincide with the presence of cirrus clouds[16–18], little is known about what happens when aircraft fly through already existing cirrus and lead to the formation of contrails within those clouds.

Compared to unperturbed cirrus regions, embedded contrails lead to a decrease in ice crystal effective radius (ICER) and an increase in ice crystal number concentration (ICNC)[19]. This manifests as an increase in cloud optical thickness (COT) that determines the cloud's net RF[20,21]. An embedded contrail becomes indistinguishable from the cloudy background within a few hours after formation as ice is removed via aggregation or sedimentation[22].

An observation-based quantification of the aviation-induced change in the optical and microphysical properties of cirrus clouds is needed to bound the climate impact of embedded contrails. While passive remote-sensing observations don't penetrate far into cirrus clouds[23,24], profiling with active instruments, such as lidar and radar, enables in-cloud observations at the flight level of a passing aircraft to assess its impact on an already existing cirrus. Earlier studies based on about 50 cases of embedded contrails over a limited area find an increase in COT[25] and ICNC[26] for perturbed cirrus regions. These observations confirm the conceptual understanding that embedded contrails are essentially contrails overlapping with existing natural cirrus or aged contrail cirrus[19] and provide valuable reference for modeling studies[22,27]. However, it is not yet clear if and how the related changes in cloud properties affect the radiative effect (RE) of the perturbed cirrus. This information is urgently needed to reduce uncertainties in the contribution of aviation-induced clouds to global warming[5,28].

This study expands the earlier work of Tesche et al.[25] in terms of both temporal and spatial coverage. The occurrence and local net RF of embedded contrails is assessed based on matching spaceborne lidar observations with aircraft position data. The work focuses on a core region that covers Europe and the North Atlantic Corridor, though near-global to global results are provided as well.

[1]Leipzig Institute for Meteorology (LIM), Leipzig University, Leipzig, Germany. [2]Department of Meteorology, University of Reading, Reading, UK.
✉e-mail: seelig@uni-leipzig.de

## Results

Figure 1 illustrates what an embedded contrail looks like in the observations used here (see "Methods" for details on the data used). The region of the cirrus that has been perturbed by the passage of an aircraft at 10.65 km height is characterized by increased extinction coefficients a few hundred meters below flight level. In contrast, lower values and a rather homogeneous structure are found further away from the aircraft's trajectory, where the cloud is assumed to be unperturbed. Vertical integration of the extinction coefficient gives COT, which peaks in the perturbed region of the cloud. The distribution of ice water content (IWC) largely follows that of COT, as IWC is inferred based on the extinction coefficient and temperature[29]. The range of ICER does not vary much over the 15-profile section, but indicates higher values in the unperturbed region of the cloud. The local net RF of the embedded contrail ($\Delta F_{net}$; see "Net radiative forcing estimation" for details) contains short- and longwave contributions and is obtained as the difference between the RE of the perturbed ($F_{net,p}$) and the unperturbed cloud ($F_{net,u}$) regions. For simplicity, net RF refers to the local net RF in the following. A value of 0.41 W m$^{-2}$ is found for the example case in Fig. 1.

An overview of the data considered here in terms of the spatial occurrence and RE of cirrus with embedded contrails is provided in Fig. 2. Matches of aircraft locations and CALIOP observations are investigated for two time periods that are constrained by the availability of waypoint data for individual aircraft (see "Spaceborne lidar data" and "Aircraft position data" for details). The first period contains global aircraft waypoint data for the year 2012 that could be matched to 19,611 cases of CALIOP observations within 5–30 min of an aircraft passing an existing cirrus cloud or a contrail cirrus above 5 km height (see "Identification of embedded contrails" for details). The second period is restricted to flights to and from Europe for four months, seasonal representatives of the years 2015–2021, and gives 21,417 such matches. The two data sets share a region of similar coverage that is used here to reconcile the observations. Incidentally, this shared region includes the North Atlantic Corridor and Europe, which were recently found to be areas in which conditions for cirrus occurrence

almost always overlap with conditions for forming long-lived contrails[18].

The occurrence of embedded contrails is closely connected to air-traffic density[4,30] with pronounced maxima over Europe, North America, and the North Atlantic Corridor. The majority of observations (62%) occur during daytime when the reflection of incoming solar radiation provides a negative contribution to $F_{net}$ that can dominate the overall effect as COT increases (Fig. 2c, d and Hong and Liu[20]). The 38% of cases found during nighttime contribute disproportionally more to the positive $F_{net}$ in line with earlier studies that consider the diurnal variation in the net RF of contrails and contrail cirrus[4,31,32]. About 82.5% of all cirrus with embedded contrails in Fig. 2 are warming. On average, a RE of the perturbed region of cirrus clouds that have been affected by an aircraft's passage (cirrus plus embedded contrail) of 9.98 W m$^{-2}$ with confidence interval [9.85, 10.11] W m$^{-2}$ is obtained.

Figure 3 showcases the findings of this work. The global annual cycle of the year 2012 (left side of Fig. 3) gives values of $\Delta F_{net}$ that vary around zero with the exception of April and June, which exceed 0.5 W m$^{-2}$. Restricting the analysis of the 2012 data set to the region shared with the 2015–2021 data set (dark red box in Fig. 2) increases the range of inferred $\Delta F_{net}$. Values for June increase even further, April falls in line with most other months, and March stands out as the month with the second-largest warming effect. While IWC and ICER of the unperturbed cloud region show a clear annual cycle, it is the difference of those properties between the perturbed and unperturbed regions of a penetrated cloud (red lines and circles in Fig. 3c, d) that primarily determines their $\Delta F_{net}$. The largest net RF is connected to increased $\Delta$IWC. In contrast, $\Delta$ICER shows no apparent connection to the annual cycle of $\Delta F_{net}$.

The Europe-centered data set from 2015 to 2021 (right side of Fig. 3) shows an increasing trend in $\Delta F_{net}$ from values below zero after March 2015 to positive values in 2019. As for the year 2012, the range of inferred $\Delta F_{net}$ increases when focusing on the shared region in Fig. 2. The IWC and ICER of the unperturbed cloud region show a clear annual cycle in line with the data for 2012. However, 2018 and 2020 show particularly weak amplitudes for IWC. The $\Delta$-variables

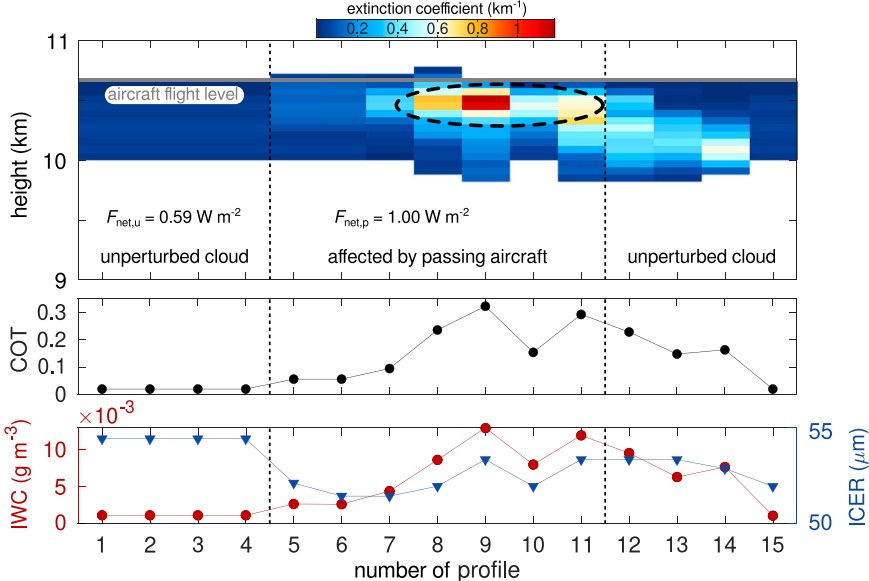

**Fig. 1 | Example of an embedded contrail and corresponding cloud properties.** Cloud-Aerosol Lidar with Orthogonal Polarization (CALIOP) extinction coefficient profiles (averaged to 5 km along-track resolution) within a cirrus cloud observed about 15 min after the passage of an aircraft (around whose location the plot is centered) at a height of 10.65 km (gray line). Vertical dashed lines separate the region in close horizontal proximity to the aircraft's location (assumed to contain the embedded contrail) to observations further away (and assumed to be unperturbed). The dashed ellipse highlights an increase in extinction coefficient attributed to the passing aircraft's effect. Numbers give the local radiative effect of the perturbed ($F_{net,p}$) and unperturbed cloud ($F_{net,u}$) regions, respectively. Lower panels show the corresponding cloud optical thickness (COT, black), ice water content (IWC, red dots), and ice crystal effective radius (ICER, blue triangles).

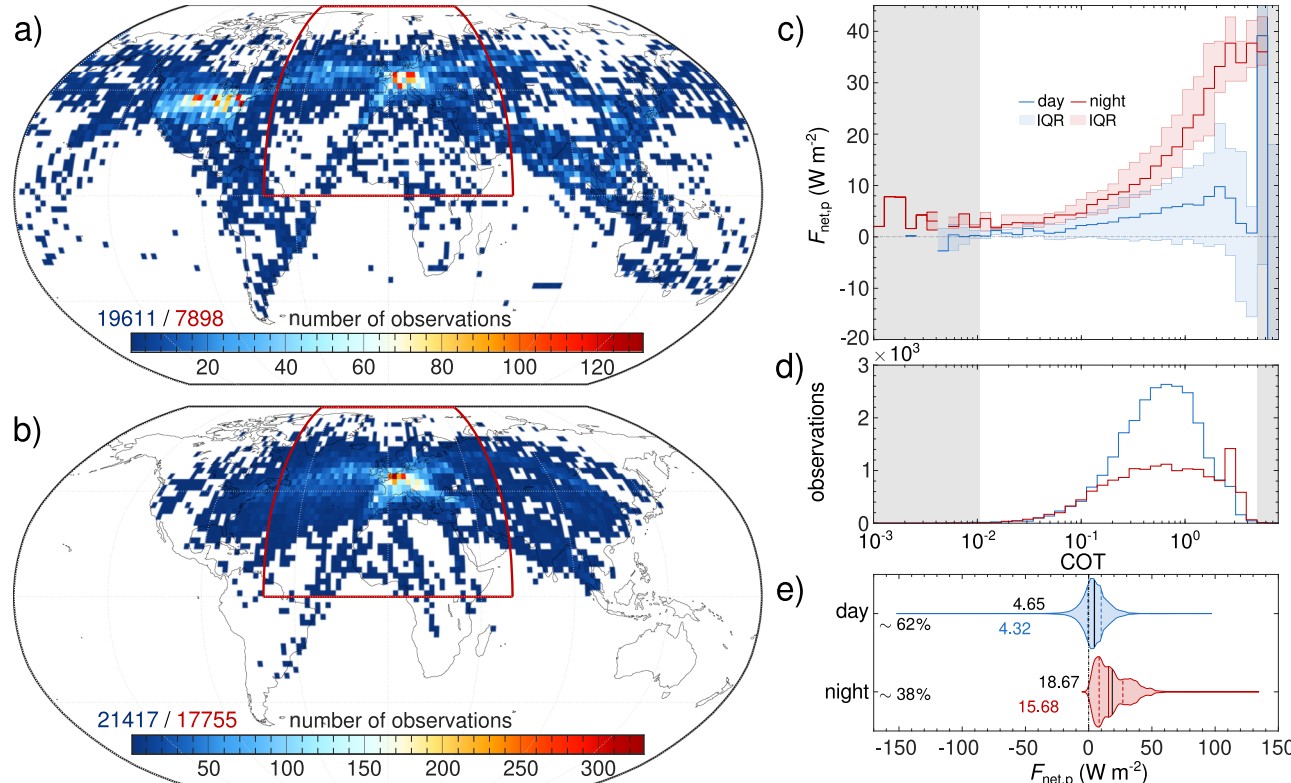

**Fig. 2 | | Spatial coverage of the considered cases and the net radiative effect (RE) of the perturbed cloud region.** The number of considered Cloud-Aerosol Lidar with Orthogonal Polarization (CALIOP) observations behind aircraft above 5 km height in grid boxes of 2.5° by 2.5° for the data sets covering the years (**a**) 2012 and (**b**) 2015–2021. For a data set comparison the shared region framed in dark red is used. The numbers above the legends indicate the number of cases of the sets found for the corresponding regions. The distributions of the mean (lines) and inter-quartile range (IQR, shaded areas) of the net RE of the perturbed region for all the observations in (**a**) and (**b**) as a function of cloud optical thickness (COT) and according to day (blue) and night (red) are shown in (**c**). The distribution of the number of observations (white area−more than 20) in the respective COT bins is shown in (**d**). The corresponding probability density distributions, including mean (black solid line and numbers), median (blue/red solid line and numbers), and inter-quartile range (blue/red dashed lines), are shown in (**e**). Percentages represent the occurrence rate of daytime and nighttime cases.

appear to stagnate until 2020. They show a larger variation starting with the COronaVIrus Disease 2019 (COVID-19) lockdown in 2020, during which aviation and its emissions were highly reduced. This is reflected in the minimum frequency of observations during June 2020. The large drop in air-traffic density lead to a strong reduction of anthropogenic perturbation at cirrus level and, thus, to a cloud background much closer to pre-industrial conditions[33]. Consequently, the largest and second-largest cooling effect is found during those periods when individual aircraft emissions of soot particles and aqueous aerosol particles were less likely to add to an already saturated aerosol particle effect. The ambient atmosphere adjusts to a state with almost no air traffic, which is also reflected in the adaptation of net RF to it.

Our findings on the $\Delta F_{net}$ of embedded contrails are summarized in Table 1. We find that embedded contrails have a warming effect of 60 to 110 mW m$^{-2}$, depending on the considered waypoint data set and time period. Focusing on the shared region gives a wide range of $\Delta F_{net}$ from clearly cooling to moderately warming. We find that embedded contrails have a cooling effect of −140 to −460 mW m$^{-2}$ during the day. However, the nighttime observations, which make up only slightly more than one third of cases, cause warming of 450 to 490 mW m$^{-2}$ and, thus, dominate the overall effect. This is in line with earlier studies of contrails and contrail cirrus that highlight the impact of the absence of shortwave cooling during night[4,31,32]. We find that two-fifths of the embedded contrails in our database have existed for less than 15 min at the time of observation. Those young contrails generally show a larger warming effect compared to persistent ones that are mostly cooling.

Together with the discussion of Fig. 3, Table 1 highlights that the $\Delta F_{net}$ of embedded contrails depends on the properties of the unperturbed background cirrus, the Sun position (time of day and location), and the time since formation. In addition, embedded contrails seem to lead to a larger magnitude in $\Delta F_{net}$ during conditions of reduced background perturbation of the atmosphere, such as during the COVID−related reduction in air traffic[30,34,35]. However, such effects seem to act on short time scales as $\Delta F_{net}$ has since returned to pre-COVID levels.

## Discussion

The largest individual positive contribution of aviation to net RF comes from contrails and contrail cirrus[1]. However, the confidence level was assessed to be low due to potential missing processes as outlined in ref. 2. A range of modeling studies has assessed the global and annual mean net RF of contrails and contrail cirrus to range between 19 and 98 mW m$^{-2}$ with a mean value of 57 mW m$^{-2}$, e.g.,[2–4,9,11,12]. In contrast, no such estimates are yet available for contrails that form in already existing cirrus clouds. Here, we investigate the occurrence and local net RF of such embedded contrails and the clouds they have perturbed shortly after the passage of aircraft, based on more than 40,000 cases in which spaceborne lidar observations are matched to the passage of an aircraft with lead times of less than 30 min. Our findings thus represent a quantitative assessment to the missing contribution to one of aviation's main non-CO$_2$ effects.

In contrast to modeling studies that compare simulations with and without contrails to assess their overall effect, our observation-based

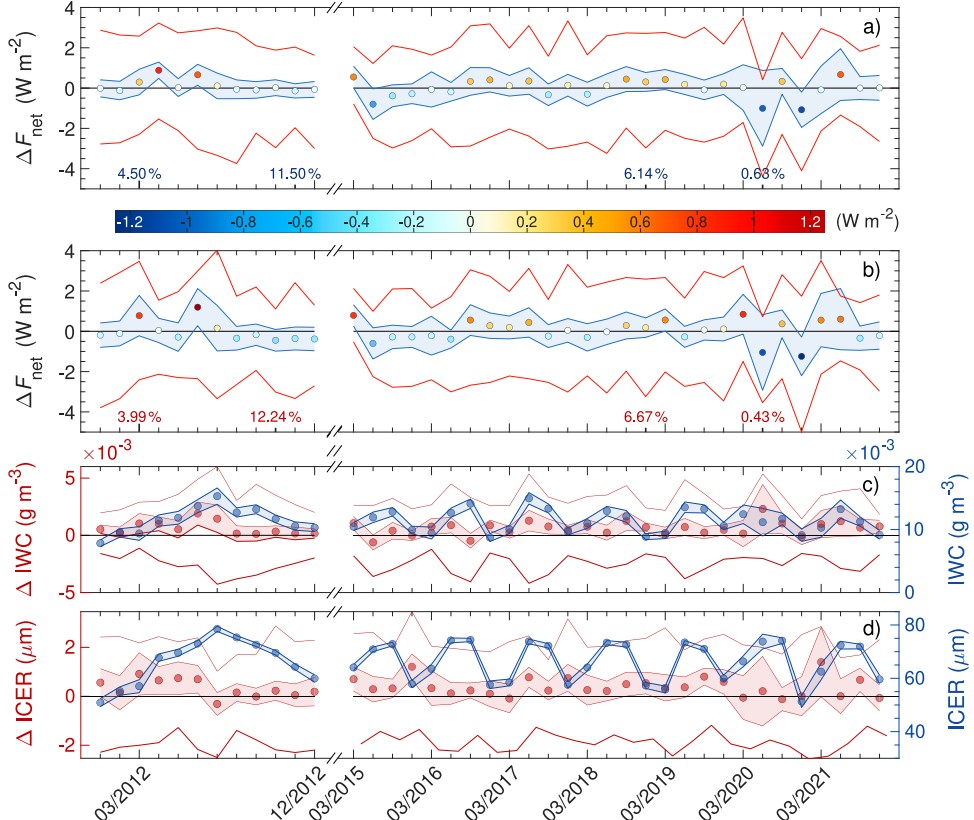

**Fig. 3 | Local net radiative forcing ($\Delta F_{net}$) of embedded contrails and corresponding cloud properties.** Monthly averages (circles), confidence interval (shaded area), and inter-quartile range (red lines) of $\Delta F_{net}$ for observations within the global (**a**) and the shared (**b**) regions in Fig. 2a, b. The bold ticks on the abscissa indicate the months, and the numbers the minimum and maximum frequency of observations according to the respective data sets. **c**, **d** as in (**b**), but for ice water content (IWC) and ice crystal effective radius (ICER) of isolated embedded contrails (red, Δ refers to the difference of perturbed and unperturbed regions) and the unperturbed cloud region (blue). For the sake of clarity, in (**a**–**d**), the baseline is shown as a horizontal solid black line.

**Table 1 | Local net radiative forcing of embedded contrails for different scenarios**

| Region | 2012 | | 2015–2021 | |
|---|---|---|---|---|
| Global | 0.11 [−0.02, 0.23] | | 0.06 [−0.06, 0.17] | |
| Shared | −0.13 [−0.32, 0.06] | | 0.05 [−0.08, 0.17] | |
| | day (61%) | night (39%) | day (63%) | night (37%) |
| Global | −0.26 [−0.40, −0.13] | 0.67 [0.42, 0.92] | −0.15 [−0.28, −0.01] | 0.40 [0.18, 0.62] |
| | day (66%) | night (34%) | day (69%) | night (31%) |
| Shared | −0.46 [−0.67, −0.25] | 0.49 [0.11, 0.87] | −0.14 [−0.28, 0.00] | 0.45 [0.19, 0.71] |
| | young (40%) | persistent (60%) | young (40%) | persistent (60%) |
| Global | 0.24 [0.04, 0.45] | 0.01 [−0.16, 0.17] | 0.14 [−0.04, 0.33] | −0.01 [−0.16, 0.15] |
| Shared | −0.05 [−0.35, 0.25] | −0.18 [−0.42, 0.06] | 0.15 [−0.05, 0.35] | −0.03 [−0.19, 0.14] |

$\Delta F_{net}$ (mean and confidence interval in W m$^{-2}$) related to the different time periods and separated according to regional coverage (global versus shared region), occurrence during day or night, and contrail age (young: $5 \leq \Delta t$ (min)<15 and persistent: $15 \leq \Delta t$ (min)<30).

analysis focuses on the local impact of individual aircraft. In this scenario, the case-specific cloud background might actually contain perturbations from earlier aircraft. Those would reduce the contrast between perturbed and unperturbed cloud regions compared to a lone-aircraft scenario. In reality, however, it is more likely that the impact of multiple aircraft flying through the same cloud scales nonlinearly and eventually saturates so that later perturbations become indistinguishable from the background. This is also reflected in our data set when inspecting subsequent cases in regions of high air-traffic density. One might thus consider our estimate of net RF as a lower estimate in absolute terms.

The occurrence of embedded contrails is regionally dependent and closely connected to air traffic density. Considering all cases from 2012 and 2015 to 2021, a positive mean net RE of the perturbed region (i.e., the combined effect of an already existing cirrus and an embedded contrail) is estimated to -10 W m$^{-2}$ and is comparable to estimates e.g., by Gasparini and Lohmann[36] (5.7 W m$^{-2}$) and Kienast-Sjögren et al.[37] (overcast sky -10 W m$^{-2}$). 83% of those cases contribute to warming although 62% of embedded contrails were observed during the day. When isolating the net RF of embedded contrails by subtracting the RE of the unperturbed cirrus region from that of the perturbed one, we identify a warming effect of 60 to 110 mW m$^{-2}$

depending on the considered aircraft position data set and the time period. Embedded contrails, which exist during the day, show a consistent cooling effect of −150 to −260 mW m$^{-2}$ caused by the solar contribution to the net RF[21]. The reduced air traffic during the night means that only 38% out of 41,028 cases of embedded contrails are found in the absence of a shortwave component to net RF. Because those cases show a net RF of 400 to 670 mW m$^{-2}$, they dominate the overall effect and shift it to net warming.

Our data set includes embedded contrails observed at 5 to 30 min after the passage of an aircraft through a cirrus cloud. This allows us to analyze the temporal evolution of their net RF. About three-fifths of embedded contrails are older than 15 min, and those persistent cases are found to exert a smaller net RF than the younger ones—likely as the result of contrail spreading, which mixes the original contrail with the ambient cloud. This is supported by a reduction in the differences in COT and IWC, and a slight growth in the difference of ICER between the perturbed and unperturbed cloud regions when contrasting young and persistent contrails. It is likely that ICER increases as long as the ambient air is supersaturated. In the longer term, atmospheric adjustments such as changes in humidity or lapse rate[30] and cirrus cloud adjustments caused by, e.g., radiation and latent heating[38] probably modify the local instantaneous RF calculated here. However, disentangling these processes would require a model capable of representing embedded contrail and cirrus evolution, which is beyond the scope of our study. As a potential next step for extending the time delay beyond 30 min, we consider using forward trajectories from the position of the aircraft to better constrain the location of the contrail in the CALIPSO curtain.

Wolf et al.[21] demonstrate that the cloud RE is most sensitive to changes in ICER and IWC and less so to changes in Sun position, surface albedo, and surface temperature. However, the cloud RE also depends on the absolute values of ICER and IWC and, therefore, their individual combinations. While this ambiguity complicates a potential parameterization of net RF to contrail-related changes in cloud microphysical properties, ICER and IWC are regarded as the main drivers of the solar and terrestrial components as well as cloud RE.

Besides a seasonal variation, an increasing trend in net RF from negative to positive is found for the time period from 2012 and 2015 to 2019. The sudden reduction in air traffic due to COVID-19 caused an immediate reduction in the emission of water vapor and ice-nucleating particles by aircraft in 2020. This lead to an atmospheric background state that was no longer saturated with respect to the impact of non-CO$_2$ effects of aviation and, likely, closer to pre-industrial conditions. As a consequence, a strong signal in the net RF of embedded contrails is found during that time period. Nevertheless, the effect was restricted to a relatively short time scale as the net RF of embedded contrails had already returned to pre-COVID levels in 2021.

While the inferred average net RF of 60 to 110 mW m$^{-2}$ (depending on considered time period) is related to global observations, it refers to the local effect at the level of individual clouds. We follow two approaches for transforming those values into an estimate of the annual global mean net RF of embedded contrails. Both try to weigh the local net RF by assuming that it can only contribute to the global mean net RF where embedded contrails are present. For a first estimate, we re-grid the data in Fig. 1a, b to 1° by 1° resolution (as 2.5° by 2.5° grid boxes overrepresent regions with low flight density) to obtain an embedded contrail fraction of around 0.10. We then scale the findings in Table 1 by this number and get values in the range from 5 to 11 mW m$^{-2}$. For a second estimate, we also re-grid the data in Fig. 1a, b to a 1° by 1° resolution and include empty grid boxes in the averaging, assuming they contribute a RF of zero to the global average. This gives a globally averaged net RF in the range from 6 to 9 mW m$^{-2}$. Knowledge of the distribution of embedded contrail cover would improve that admittedly crude estimate, but it provides an observational constraint on a previously unquantified non-CO$_2$ effect of aviation. Overall, our

results suggest that the climate effect of embedded contrails is on the order of 10% of the effect of line-shaped contrails[2] and, thus, a non-negligible contributor to the climate impact of aviation.

## Methods

### Spaceborne lidar data

Cirrus information considered in this study is taken from observations of the Cloud-Aerosol Lidar with Orthogonal Polarization (CALIOP) on the Cloud-Aerosol Lidar and Infrared Pathfinder Satellite Observations (CALIPSO) satellite[39]. CALIOP is a depolarization lidar that emits linearly polarized laser light at 532 and 1064 nm wavelengths. Back-scattering is measured at 532 nm wavelengths, discriminating for parallel and perpendicular polarization, and at 1064 nm wavelength. The instrument has been in operation from June 2006 to August 2023. Here, we use the level 2 5-km cloud profile product (version 4.20 until June 2020 and version 4.21 from September 2020). Data are provided with a horizontal resolution of 5 km in height bins of 60 m at cirrus level. The parameters used are the time and the location of observation, underlying surface type (IGBP_Surface_Type), solar zenith angle (SZA) threshold (Day_Night_Flag), 532 nm extinction coefficient, cloud-and-aerosol discrimination (CAD) score, and vertical feature mask (VFM). CALIPSO files also contain information on temperature and relative humidity from the MERRA-2 reanalysis, as well as IWC retrieved from a parameterization using the MERRA-2 temperature ($T_{MERRA-2}$)[29]. IWC includes a parameterization for ICER for three different temperature ranges[40]. The temperature was also used to calculate saturation vapor pressure (SVP) over water and ice following Sonntag[41]. Afterwards, SVP and relative humidity were used to calculate ice supersaturation (ISS).

### Aircraft position data

Two aircraft waypoint data sets that provide information on the position, height, and timing of individual aircraft are used to collocate CALIOP observations with the passage of commercial aircraft. The first observation period, covering the year 2012, is a composite of global aviation data from the U.S. Department of Transportation (DOT) Volpe Center, the U.S. Federal Aviation Administration (FAA), and EURO-CONTROL (European Organisation for the Safety of Air Navigation) and has been previously used, e.g., by Duda et al.[10]. The second observation period from 2015 to 2021 is covered by the EURO-CONTROL R&D data set but is restricted to flights originating from or going to Europe for the months March, June, September, and December. These months are regarded as seasonal representatives.

### Identification of embedded contrails

This work follows the approach introduced in Tesche et al.[25], and we refer readers to that paper and its Supplementary Material regarding further details not outlined below, including the role of the observational geometry. Intercepts between the CALIPSO ground track and the flight tracks of individual aircraft above 5 km height are identified using the TrackMatcher tool[42,43]. We only consider cases with CALIOP observations within 5 to 30 min after an aircraft's passage. Aircraft lead times shorter than 5 min are excluded as contrails are still in their formation stage during that time[5,44]. We extract the 15 lidar profiles around the location of the aircraft-CALIPSO intercept from the CALIPSO level 2 cloud profile product. For quality assurance, we require extinction coefficients in individual profiles to be flagged as cirrus in the VFM and to show a CAD score ≥ 20 to reliably identify clouds. The thus selected data have to correspond to horizontally contiguous cloud layers that contain the flight level of the aircraft of the corresponding intercept, as well as an area of at least 250 m below flight level, which is where the embedded contrail would sediment to. The COT, cloud geometrical thickness, and IWC are calculated according to the cloud profiles that meet the criteria above. COT of unperturbed cirrus clouds does not follow a Gaussian distribution.

Hence, normalized COT is obtained by normalization of COT data points of individual intercepts with respect to the respective maximum COT value, as in Tesche et al.[25] for comparability of the different cases. Each case of 15 profiles is then split into a 7-profile section around the aircraft track and two 4-profile sections further away on either side of the aircraft track. The two regions are assumed to represent cirrus that is perturbed or unperturbed, respectively, by the passing aircraft, as shown in Fig. 1. The impact of the embedded contrail is then quantified by contrasting the parameters corresponding to the perturbed and unperturbed regions of the observed cloud.

### Net radiative forcing estimation

The estimation of the net RF of embedded contrails is based on the look-up tables (LUTs) of Wolf et al.[21]. Details of and simplifications in those calculations can be found therein. Differences related to meteorological conditions used in the LUTs and for the observations cancel out in first approximation as we focus on the difference between the radiative calculations with and without embedded contrails.

The LUTs have eight dimensions and require information on SZA at the time and location of observation; IWC, ICER, ice crystal shape, and temperature of the cirrus cloud ($T_c$); surface temperature ($T_{srf}$); COT of a potential liquid cloud below cirrus ($COT_l$); and surface albedo ($\alpha_{srf}$). In our case, IWC, ICER, and $COT_l$ are obtained as median values with respect to the perturbed and unperturbed regions of the observed cloud (see "Identification of embedded contrails"). SZA, $T_c$, $T_{srf}$, $\alpha_{srf}$, and ice crystal shape are obtained for the intercept points between CALIPSO ground track and the flight paths and, for simplicity, are used for both the perturbed and unperturbed regions.

Profile-wise $COT_l$ is retrieved as vertical integral of those extinction-coefficient values that show a non-cirrus flag in the VFM and a CAD score $\geq 20$ before medians are calculated for the two cloud regions. Wolf et al.[21] implemented liquid clouds in radiative transfer (RT) simulations as stratiform low-level clouds with an average cloud top height of 1500 m. However, the calculation of $COT_l$ also includes profiles exceeding cloud tops of 1500 m, which might be a source of uncertainty in net RF. In the absence of liquid clouds, $COT_l$ is set to zero. The solar position algorithm for solar radiation applications[45] is used to calculate SZA. $T_c$ and $T_{srf}$ represent the temperature at flight level and at the surface, respectively. For the selection of ice crystal shape, we use the habit diagram of Bailey and Hallett[46] (Fig. 5 therein). Based on $T_c$ and ISS (see "Spaceborne lidar data"), ice crystals are classified as droxtals ($-0.05 \leq ISS < 0.05$), aggregates ($T_c \leq -40\,°C$ and ISS outside the droxtal range), or plates ($T_c > -40\,°C$ and ISS outside the droxtal range). $\alpha_{srf}$ is selected according to surface type from Clouds and the Earth's Radiant Energy System (CERES) Surface and Atmospheric Radiation Budget (SARB) scene identification values based on[47] (https://ceres.larc.nasa.gov/data/general-product-info/#ceres-nested-10-processing-grid, last access: 8 May 2025).

Once all parameters are retrieved, they are mapped to the resolution provided by Wolf et al.[21] (Table 4 therein). After the cloud RE with its solar and terrestrial contributions is obtained for the perturbed ($F_{net,p}$) and the unperturbed ($F_{net,u}$) cirrus regions, the local net RF of an embedded contrail is obtained as

$$\Delta F_{net} = F_{net,p} - F_{net,u}$$

Based on the CALIPSO day-night flag (all the profile flags corresponding to the perturbed or unperturbed region must fulfill either day or night), cases are grouped into day and night. For a temporal assessment of $\Delta F_{net}$, cases are sorted according to two groups as contrail age[5]. Cases with aircraft lead times of less than 15 min are defined as young embedded contrails in the vortex stage. Cases with larger aircraft lead times up to the considered maximum of 30 min are defined as persistent embedded contrails that are in transition to or already in the spreading stage.

### Uncertainties in the estimate of net RF

The LUTs used to estimate the net RF of embedded contrails[21,48] were calculated by varying SZA, IWC, ICER, $T_c$, $T_{srf}$, $\alpha_{srf}$, atmosphere profile, $COT_l$, and ice crystal shape. The corresponding parameter ranges were chosen based on typical values from observations[49,50].

While the LUT's parameter ranges were carefully chosen, they may not be sufficient to cover the range of, e.g., $T_c$, $T_{srf}$, IWC, and ICER obtained from the MERRA-2 reanalysis. As a consequence, input values outside the covered parameter range are set to give the net RF related to the smallest or largest possible LUT value.

The cloud RE from the LUT is neither extrapolated nor interpolated between bins, since the relationships between input parameters and cloud RE are non-linear. This coarse binning may partially mask uncertainties in estimating net RF related to $T_{MERRA-2}$, IWC, and ICER because variations in those parameters can only lead to the selection of an incorrect net RF estimate for values close to a LUT bin edge.

Sensitivity tests show that the cloud RE is controlled primarily by ICER and IWC[21]. Both parameters depend on temperature, which is why the uncertainty in the $T_{MERRA-2}$ profile is expected to have the largest impact on our estimates of the net RF. To quantify the related uncertainty, we vary $T_{MERRA-2}$ by $\pm 1\,K$ in the input of $T_c$, $T_{srf}$, and the parameterization of ICER, while IWC is not modified. The resulting mean local net RF shows a variance of $\pm 0.09\,W\,m^{-2}$ and is within the range of the confidence interval in Table 1.

The cloud RE in the solar wavelength range is further affected by SZA, which can be accurately calculated, and $\alpha_{srf}$. Surface type can affect the surface net longwave flux of up to $6\,W\,m^{-2}$ [47], which corresponds to an uncertainty of 4% in broadband albedo. The terrestrial contribution depends primarily on $T_c$. Assuming thermal equilibrium between cirrus and surrounding atmosphere, $T_c$ is equal to $T_{MERRA-2}$ at the same level.

Finally, systematic biases in cloud RE may also be caused by the assumption of plane-parallel clouds in the one-dimensional (1D) RT calculations used to create the LUTs. While this assumption may be suited for aged and stratiform contrails, it is violated in case of younger and line-shaped contrails. However, 1D and 3D RT calculations of cirrus and contrails are consistent over a wide range of conditions, with the largest differences for large SZA, i.e., during sunrise and sunset[51–53]. The observations used in our study occur rarely, less than 5%, near sunrise and sunset.

## Data availability

The EUROCONTROL R&D data set is openly available at https://www.eurocontrol.int/dashboard/rnd-data-archive. The data sets containing the cloud radiative forcing for the three individual ice crystal shapes are available in the *Zenodo* platform at https://doi.org/10.5281/zenodo.8159286[48]. The data that support the findings of this study are openly available in the *Zenodo* platform at https://doi.org/10.5281/zenodo.15600380[54].

## Code availability

The original TrackMatcher package[42,55] is available at https://github.com/LIM-AeroCloud/TrackMatcher.jl under the GNU General Public License v3.0 or higher. The version used in this study including updates to handle the EUROCONTROL R&D data set is openly available in the *Zenodo* platform https://doi.org/10.5281/zenodo.17369750.

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

## Acknowledgements

The research work for this article is part of the "Advancing mEasures to Reduce aviatiOn imPact on cLimate and enhAnce resilieNce to climate-changE" (AEROPLANE) project, which is supported by the Single European Sky ATM Research (SESAR) 3 Joint Undertaking (JU) and its founding members over GA nr. 101114682. We thank Gregg G. Fleming, of the Volpe Center, for providing the aircraft waypoint data set for 2012, supported by the Open Access Publishing Fund of Leipzig University.

## Author contributions

MT and TS conceived the idea for this study and wrote the initial manuscript. TS performed the data analysis and prepared the figures with feedback from MT. KW provided support in the application of the radiative forcing look-up table. KW, NB, MT, and TS contributed to the interpretation and discussion of the findings and the revision of the manuscript.

## Funding

## Competing interests

The authors declare no competing interests.
