## [Transparent Peer Review file · Nature Communications]

Quantification of the Radiative Forcing of Contrails Embedded in Cirrus Clouds

Corresponding Author: Dr Torsten Seelig

Version 0:

Reviewer comments:

Reviewer #1

(Remarks to the Author)

The manuscript takes on the challenge of understanding and quantifying the local radiative forcing resulting from contrails embedded in pre-existing cirrus. The authors achieve this goal without relying on contrail or weather modelling, instead relying on LiDAR to collocate perturbed cirrus clouds with flight tracks shortly after the passage of commercial aircraft. Using a high quality look up table for radiative forcing calculations they find that, much as with conventional contrails, the net effect is warming but that there is strong dependence on context (including time of day).

Quantifying a new component of contrail-cirrus radiative forcing without resorting to atmospheric modelling is a useful calculation, and provides an important advance towards an interesting and important question. However, the manuscript – likely inadvertently – overstates the conclusions, seeming to provide an estimate of local radiative forcing but giving the impression that a new component in aviation’s global, annual mean RF has been calculated. Some additional lack of clarity regarding the methods used adds to this confusion. I have elaborated more on this in my comments below, but until these issues are addressed I would not recommend this manuscript for publication. If however the authors can carefully caveat the work and make this distinction clearer, as well as addressing the methodological concerns below, I believe that the content of this paper is of interest and importance to the aviation climate impacts community. Since however we do not yet know the area over which cirrus is affected and therefore cannot produce an estimate of the global-scale effect on aviation’s RF, this remains an incremental advance which mostly extends an existing technique to a larger dataset.

The most significant concern regards the framing of the paper. The abstract refers to calculation of “the net radiative forcing (RF) of embedded contrails”, providing an “annual mean net RF” of 60 mW/m². On first read, this is an astonishing outcome – almost double the net radiative forcing reported by Lee et al. (2021) for aviation CO₂ in 2018. However, until the very last line of the conclusions I found that it was very unclear whether the manuscript is referring to the forcing within the affected area (i.e. only over the area directly affected by the passage of aircraft), or if this is a global mean. The fact that this refers specifically to the forcing specifically in a region around the affected cloud, rather than a global, area-averaged value, is not clear from either the main text or from the methodology sections. I would recommend that the authors provide clear and specific statements, in the abstract, introduction, and results, which state that the results here correspond only to the average local radiative forcing in cloud which has been recently modified by the passage of aircraft, and that this is not an estimate of the global, annual mean radiative forcing associated with embedded cirrus.

A related issue regards the degree to which these estimates can be considered to reflect the full impact of embedded contrails. As I understand it the results here cover only a 25 minute period (from 5 to 30 minutes) following an aircraft’s passage. This is a sensible restriction given the cumulative difficulty of identifying aircraft flight tracks at longer and longer times since a contrail’s creation, but it inherently limits the scope of the study. I would ask that the authors comment on the potential that longer term impacts might dominate the highly localized RF calculated here. In particular, given that cirrus clouds typically have lifetimes on the order of tens of hours (e.g. Dobbie et al., 2001), it would be useful to understand how these results can or cannot be extrapolated to understand the overall RF effect of embedded contrails when considering the full lifetime.

Secondly, I am concerned about the method used to define the affected region. Section 4 states that “each case of 15 profiles is then split into a 7-profile section around the aircraft track and two 4-profile section further away on either side of the aircraft track. The two regions are assumed to represent cirrus that is perturbed or unperturbed, respectively”. It is also

stated that the CALIOP track resolution is 5 km. This implies that the RF is always calculated based on the degree to which a 35 km-wide area is changed. This makes sense given the somewhat inevitable imprecision in locating the flight track, especially after 30 minutes of advection. However it also means that all calculations are really of the average RF effect over a 35 km region around an aircraft. This is likely to be 5-10 times the actual affected region, given that a typical vortex system behind an aircraft might spread emissions over a vertical depth of <500 m (Unterstrasser 2016) and even a consistent, strong vertical wind shear of 5 m/s/km would only spread these emissions by ~5 km over 30 minutes. This in turn implies that the truly "local" RF is 5-10 times greater. I recommend that the authors state explicitly how this horizontal region is calculated, how it is used in calculating the average RF, and the implications for the estimated mean net RF.

I would also like some additional details to address concerns regarding the accuracy of the technique. First, it seems that the authors are assuming negligible motion of the aircraft emissions after passage of the aircraft, even up to 30 minutes later. I would recommend including a quantitative analysis of the degree to which this assumption (if present) may propagate to errors in the estimated RF effect (or lack thereof) of embedded contrails. This is particularly concerning given that the estimated RF appears to fall over time; how can one be certain that this is due to changes in the cirrus, as opposed to being because of misalignment between the observation and the affected cloud?

References

Dobbie, Steven, and Peter Jonas. "Radiative influences on the structure and lifetime of cirrus clouds." *Quarterly Journal of the Royal Meteorological Society* 127.578 (2001): 2663-2682.

Lee, David S., et al. "The contribution of global aviation to anthropogenic climate forcing for 2000 to 2018." *Atmospheric environment* 244 (2021): 117834.

Unterstrasser, Simon. "Properties of young contrails—a parametrisation based on large-eddy simulations." *Atmospheric Chemistry and Physics* 16.4 (2016): 2059-2082.

Reviewer #2

(Remarks to the Author)

1. Noteworthy Results

The manuscript presents the first large-scale, observation-based quantification of the net radiative forcing (RF) of contrails embedded within cirrus clouds, using a dataset of approximately 40,000 cases.

The study finds that embedded contrails have a predominantly warming effect, with a mean net RF of approximately 60 mW m⁻², and that this effect is especially pronounced at night.

The research demonstrates that the radiative impact of embedded contrails can be of similar magnitude to that of traditional line-shaped contrails and contrail cirrus, highlighting an important and previously underappreciated component of aviation's climate impact.

The analysis also captures the reduction in RF during the Covid-19 pandemic, providing a unique real-world test of the sensitivity of atmospheric RF to changes in air traffic.

2. Significance to the Field

This work is highly significant for the atmospheric science and climate modeling communities, as it fills a notable gap in the quantification of aviation-induced cloud effects.

The findings will be of interest not only to those studying aviation impacts, but also to researchers focused on cloud microphysics, remote sensing, and climate mitigation strategies.

Compared to the established literature (e.g., Burkhardt & Kärcher, 2011; Lee et al., 2021; Singh et al., 2024), this study provides a novel observational perspective and extends previous modeling and limited-case studies to a much larger and more representative dataset.

The work is original and does not duplicate existing studies.

3. Support for Conclusions

The conclusions are generally well supported by the data and analysis presented.

The use of both perturbed and unperturbed cloud regions for comparison is a strength, as it allows for a direct estimate of the incremental effect of embedded contrails.

The claims about the significance of embedded contrails to overall aviation RF are justified; however, the extrapolation from local to global effects is acknowledged as a limitation and could be discussed further.

4. Data Analysis, Interpretation, and Conclusions

The data analysis is thorough and uses appropriate statistical methods.

The interpretation of the results is logical and consistent with the data.

There are no major flaws that would prohibit publication, but the authors should clarify the limitations associated with the look-up table approach and the potential impact of one-dimensional radiative transfer assumptions.

The discussion of uncertainties is adequate, though a more explicit comparison with the uncertainties in previous studies would be beneficial.

5. Methodology

The methodology is sound and meets the standards expected in the field.

The use of spaceborne lidar, combined with aircraft position data and robust radiative transfer modeling, is appropriate and innovative.

The criteria for identifying embedded contrails and the temporal/spatial matching are clearly explained.

6. Reproducibility

Sufficient detail is provided in the methods section to allow for reproduction of the study.

The authors have made their data and code openly available, which further enhances reproducibility.

Minor improvements could be made by including more explicit instructions or flowcharts for the data processing steps.

7. Additional Comments and Suggestions

Consider citing the recent review by Singh et al. (2024, ACP 24, 9219–9300 <https://acp.copernicus.org/articles/24/9219/2024/>) to further contextualize the results within the broader field.

The manuscript is generally well written, but some sections could be streamlined for clarity.

The figures are informative, but captions could be expanded for standalone comprehension.

The limitations and future work sections could be strengthened by discussing how the approach might be adapted for global, area-averaged RF estimates.

Recommendation:

The manuscript represents a significant and original contribution to the field. With minor revisions to address the points above, I recommend it for publication.

Reviewer #3

(Remarks to the Author)

General comment:

This manuscript presents a timely and important study on the climate impact of embedded contrails—those forming within pre-existing cirrus clouds. By combining aircraft position data with satellite lidar observations, the authors compile a substantial dataset to estimate the previously underexplored net radiative forcing (RF) of these contrails using radiative transfer modeling. The analysis reveals that embedded contrails have an overwhelmingly warming effect, with younger contrails contributing more strongly to warming, while persistent ones tend to have a cooling impact. The study offers meaningful insights for aviation induced climate impact assessments, addresses a gap in current understanding, and is methodologically sound. The manuscript is well-written, and I recommend publication after minor revisions.

Major comment:

1. In Sect. 2, regarding the selection of unperturbed cirrus. The description of “lower values and a rather homogeneous structure are found further away from the aircraft’s trajectory” raises a question — could these regions represent older, sedimented contrails? If such cases are excluded from the statistics, and given that they tend to have lower or cooling RF as discussed later, their omission may lead to an overestimation of the overall RF.
2. In Sect. 3, the statement that “the net RF is controlled primarily by ICER as this parameter affects both the solar and the thermal infrared components” could benefit from further clarification. If I understand correctly—particularly in reference to the Nakajima and Nakajima (1999) bi-spectral method used to retrieve optical depth and effective radius—ICER mainly influences the thermal infrared component, while its impact on the solar component is more limited.
3. In Sect. 4.4, the estimation of the net RF of embedded contrails is based on look-up tables (LUTs). Could you clarify the assumptions or settings used for temperature, humidity, and overall atmospheric profiles in these LUTs?

Minor comment:

1. In Sect. 2, when introducing the net RF, the phrase “The net RF of the embedded contrail (ΔF_{net})” should include a link to the Methods section, where the calculation or definition is provided.
2. In Sect. 2, the manuscript states that global matches between aircraft waypoints and CALIOP observations for 2012 are selected as the first period, while the second period focuses on flights to and from Europe from 2015 to 2021. It would be helpful to clarify in the main text why 2012 was chosen as the reference year and why this period covers global data, whereas subsequent years are geographically limited to Europe. Although these details are partly provided in the Methods, a brief explanation in the main text would avoid the abrupt impression.
3. In Sect. 3, “besides a seasonal variation, an increasing trend in net RF from negative to positive is found for the time

period from 2012 to 2019". Changing to "2012 and 2015 to 2019" would be more correct.

Version 1:

Reviewer comments:

Reviewer #1

(Remarks to the Author)

Since this is a second review, I have restricted my comments to concerns which were raised during the first round.

I would like to thank the authors for their careful and comprehensive response to my previous comments. The updated abstract is dramatically improved, and my concerns regarding local vs global warming have been fully addressed. I also consider their response regarding the methodological questions adequate.

However, I would request one additional change. I was a little confused by the response regarding longer-term changes (i.e. those which do not manifest within 30 minutes of an aircraft's passage), which discusses only the possibility of changes in humidity/lapse rate and cirrus cloud adjustments and states that "disentangling these processes would require a model capable of representing embedded contrail and cirrus evolution". However, the later author response stated that the time delay was kept below 30 minutes because "this is the time period for which even a strongly advected contrail should still be within the perturbed section of the CALIPSO track". These responses seem inconsistent, and knowing why this 30 minute limit exists could be important to the reader. Specifically, I would be grateful if the authors could clarify the degree to which the 30 minute limit is dominated by wind speed error (i.e. the inability to accurately calculate flight track advection), the width of the CALIPSO track, and the lack of an accurate model of contrail/cirrus evolution. This could help to prioritize research, given that each of these issues is resolvable but with very different approaches. Since this is mostly a clarification I would consider this a minor change, but one I hope that the authors will consider as it would help increase both the readability and direct utility of the manuscript.

Reviewer #3

(Remarks to the Author)

I appreciate the authors' efforts in improving the careful interpretation of the annual mean net RF of embedded contrails.

I have only one remaining question regarding the use of an embedded contrail fraction of around 0.10. This appears to be a significant scaling factor for the global net RF estimate. The authors state:

"For a first estimate, we re-grid the data in Figure 1a and b to 1° by 1° resolution (as 2.5° by 2.5° grid boxes overrepresent regions with low flight density) to obtain an embedded contrail fraction of around 0.10."

Are there any references or justifications supporting the choice of a 1° by 1° resolution for this regriding step?

With this minor revision, I would support the manuscript's acceptance.

We thank the three reviewers for their detailed study of the manuscript and for providing constructive suggestions for improving the clarity and message of the paper. Please find our detailed replies (blue text) and changes to the text (blue italic) below:

REVIEWER COMMENTS

Reviewer #1 (Remarks to the Author):

The manuscript takes on the challenge of understanding and quantifying the local radiative forcing resulting from contrails embedded in pre-existing cirrus. The authors achieve this goal without relying on contrail or weather modelling, instead relying on LiDAR to collocate perturbed cirrus clouds with flight tracks shortly after the passage of commercial aircraft. Using a high quality look up table for radiative forcing calculations they find that, much as with conventional contrails, the net effect is warming but that there is strong dependence on context (including time of day).

Quantifying a new component of contrail-cirrus radiative forcing without resorting to atmospheric modelling is a useful calculation, and provides an important advance towards an interesting and important question. However, the manuscript – likely inadvertently – overstates the conclusions, seeming to provide an estimate of local radiative forcing but giving the impression that a new component in aviation’s global, annual mean RF has been calculated. Some additional lack of clarity regarding the methods used adds to this confusion. I have elaborated more on this in my comments below, but until these issues are addressed, I would not recommend this manuscript for publication. If, however, the authors can carefully caveat the work and make this distinction clearer, as well as addressing the methodological concerns below, I believe that the content of this paper is of interest and importance to the aviation climate impacts community. Since however we do not yet know the area over which cirrus is affected and therefore cannot produce an estimate of the global-scale effect on aviation's RF, this remains an incremental advance which mostly extends an existing technique to a larger dataset.

We thank the reviewer for this critical assessment of our work. We have carefully revised the text to clarify that the numbers in the results section refer to the mean local effect. In addition, we have extended the discussion to provide an informed estimate of the likely range of the global mean annual effect. We believe that the refined estimates provide a meaningful addition to the literature on aviation’s effect on climate. This is also reflected in the second half of the revised abstract (please see replies to the specific comments below for more details):

“The annual mean local net RF of individual embedded contrails ranges between -320 mW m^{-2} (2020, Covid lockdown) and 160 mW m^{-2} . Considering all cases from 2015 to 2021, we find an annual mean local warming effect of 60 mW m^{-2} . The expansion of our findings to the global scale suggests an annual global mean net RF of embedded contrails on the order of 5 mW m^{-2} . This corresponds to around 10% of the current estimate of the climate impact of line-shaped contrails and, together with recent findings that conditions for contrail formation are found most often in already-existing cirrus, suggests that embedded contrails are a non-negligible contributor to aviation’s impact on climate.”

The most significant concern regards the framing of the paper. The abstract refers to calculation of “the net radiative forcing (RF) of embedded contrails”, providing an “annual mean net RF” of 60 mW/m^2 . On first read, this is an astonishing outcome – almost double the net radiative forcing reported by Lee et al. (2021) for aviation CO₂ in 2018. However, until the very last line of the conclusions I found that it was very unclear whether the manuscript is referring to the forcing within the affected area (i.e. only over the area directly affected by the passage of aircraft), or if this is a global mean. The fact that this refers specifically to the forcing specifically in a region around the affected cloud, rather than a global, area-averaged value, is not clear from either the main text or

from the methodology sections. I would recommend that the authors provide clear and specific statements, in the abstract, introduction, and results, which state that the results here correspond only to the average local radiative forcing in cloud which has been recently modified by the passage of aircraft, and that this is not an estimate of the global, annual mean radiative forcing associated with embedded cirrus.

We thank the reviewer for pointing out this ambiguity. Indeed, the estimates provided in the abstract and in the results section refer to the local net RF, i.e., the area directly affected by the passage of aircraft. We have carefully screened those parts of the manuscript to clarify the used language.

In addition to the local net RF, we now provide an estimate of the global net RF as described in the final paragraph of the revised discussion:

“While the inferred average net RF of 60 to 110 mW m⁻² (depending on considered time period) is related to global observations, it refers to the local effect at the level of individual clouds. We follow two approaches for transforming those values into an estimate of the annual global mean net RF of embedded contrails. Both try to weigh the local net RF by assuming that it can only contribute to the global mean net RF where embedded contrails are present. For a first estimate, we re-grid the data in Figure 1a and b to 1° by 1° resolution (as 2.5° by 2.5° grid boxes overrepresent regions with low flight density) to obtain an embedded contrail fraction of around 0.10. We then scale the findings in Table 1 by this number and get values in the range from 5 to 11 mW m⁻². For a second estimate, we include empty grid boxes in Figure 1a and b in the averaging, assuming they contribute a RF of zero to the global average. This gives a globally averaged net RF in the range from 4 to 8 mW m⁻². Knowledge of the distribution of embedded contrail cover would improve that admittedly crude estimate, but it provides a first observational constraint on a previously unquantified non-CO₂ effect of aviation. Overall, our results suggest that the climate effect of embedded contrails is on the order of 10% of the effect of line-shaped contrails (Lee et al, 2021) and, thus, a non-negligible contributor to the climate impact of aviation.”

A related issue regards the degree to which these estimates can be considered to reflect the full impact of embedded contrails. As I understand it the results here cover only a 25 minute period (from 5 to 30 minutes) following an aircraft’s passage. This is a sensible restriction given the cumulative difficulty of identifying aircraft flight tracks at longer and longer times since a contrail’s creation, but it inherently limits the scope of the study. I would ask that the authors comment on the potential that longer term impacts might dominate the highly localized RF calculated here. In particular, given that cirrus clouds typically have lifetimes on the order of tens of hours (e.g. Dobbie et al., 2001), it would be useful to understand how these results can or cannot be extrapolated to understand the overall RF effect of embedded contrails when considering the full lifetime.

Longer-term adjustments to cirrus cloud lifetime and cloudiness probably play an important role in local net RF. We thank the reviewer for pointing this out. In the long term, besides natural cirrus responses to radiation, atmospheric adjustments might impact the homogeneity and lifetime. Therefore, it is difficult to extrapolate the results to understand the overall effect on net RF. However, we added the following sentence to the discussion in Section 3 to point out potential long-term effects of cloud perturbations related to embedded contrails:

“In the longer term, atmospheric adjustments such as changes in humidity or lapse rate (Teoh et al, 2024b) and cirrus cloud adjustments caused by, e.g., radiation and latent heating (Dobbie and Jonas, 2001) probably modify the local instantaneous RF calculated here. However, disentangling these processes would require a model capable of representing embedded contrail and cirrus evolution, which is beyond the scope of our study.”

Secondly, I am concerned about the method used to define the affected region. Section 4 states that “each case of 15 profiles is then split into a 7-profile section around the aircraft track and two 4-profile section further away on either side of the aircraft track. The two regions are assumed to represent cirrus that is perturbed or unperturbed, respectively”. It is also stated that the CALIOP track resolution is 5 km. This implies that the RF is always calculated based on the degree to which a 35 km-wide area is changed. This makes sense given the somewhat inevitable imprecision in locating the flight track, especially after 30 minutes of advection. However, it also means that all calculations are really of the average RF effect over a 35 km region around an aircraft. This is likely to be 5-10 times the actual affected region, given that a typical vortex system behind an aircraft might spread emissions over a vertical depth of <500 m (Unterstrasser 2016) and even a consistent, strong vertical wind shear of 5 m/s/km would only spread these emissions by ~5 km over 30 minutes. This in turn implies that the truly “local” RF is 5-10 times greater. I recommend that the authors state explicitly how this horizontal region is calculated, how it is used in calculating the average RF, and the implications for the estimated mean net RF.

The reviewer raises the issue that the features we want to detect are likely much smaller than the distance along which observations are available and, thus, might be reduced to some degree as a result of the averaging in the analysis method. This reasoning assumes that we will always find intercepts that represent a cross section through the embedded contrail that is perfectly perpendicular to the flight track, i.e., CALIPSO along-track distance is identical to distance from the aircraft’s track. Our cases, however, represent many different geometrical alignments – a problem that is already addressed in the Supplementary Material to *Tesche et al.* (2016) in which our method was first presented. For instance, wind speed and the angle between wind direction and the aircraft’s heading determine the advection of the contrail while the angle between the CALIPSO ground track and the flight track of individual aircraft determines the actual distance of an observation from that of the aircraft. These circumstances together with the noise of spaceborne lidar data complicate the scenario described by the reviewer and we carefully designed our analysis method to be consistently applicable to CALIPSO observations.

Nevertheless, the reviewer is correct that the RF of a sharply defined contrail that is only present in one of the profiles of the perturbed section of the CALIPSO intercept will be reduced by averaging it with six unaffected profiles. However, the intrinsic heterogeneity of cirrus clouds together with the specifics of the analysis procedure (signals have to relate to feature detection at 5-km horizontal averaging, *Tesche et al.*, 2016) and the intercept geometry described above indicate that the idealized scenario described by the reviewer is unlikely to dominate our data set of around 40,000 observations.

We address the reviewer’s concern by pointing the readers to our earlier work – specifically towards the additional information in the Supplementary Material – in the revised beginning of Section 4.3: “*This work follows the approach introduced in Tesche et al. (2016) and we refer readers to that paper and its Supplementary Material regarding further details not outlined below, including the role of the observational geometry.*”

I would also like some additional details to address concerns regarding the accuracy of the technique. First, it seems that the authors are assuming negligible motion of the aircraft emissions after passage of the aircraft, even up to 30 minutes later. I would recommend including a quantitative analysis of the degree to which this assumption (if present) may propagate to errors in the estimated RF effect (or lack thereof) of embedded contrails. This is particularly concerning given that the estimated RF appears to fall over time; how can one be certain that this is due to changes in the cirrus, as opposed to being because of misalignment between the observation and the affected cloud?

We understand the concern regarding the accuracy of the technique. However, we want to clarify that we are not assuming that aircraft emissions are stationary. In fact, this is the main reason why we define perturbed and unperturbed sections of the CALIPSO track. We have provided details on this in the reply to the previous comment. Here, we would like to re-iterate that we keep the time delay between aircraft and CALIPSO overpass below 30 minutes because this is the time period for which even a strongly advected contrail should still be within the perturbed section of the CALIPSO track as outlined in the earlier reply in which we address the geometry of the problem. Details on this are also provided in the Supplementary Material to *Tesche et al.* (2016) and we have added specific reference to that extra information.

References

Dobbie, Steven, and Peter Jonas. "Radiative influences on the structure and lifetime of cirrus clouds." *Quarterly Journal of the Royal Meteorological Society* 127.578 (2001): 2663-2682.

Lee, David S., et al. "The contribution of global aviation to anthropogenic climate forcing for 2000 to 2018." *Atmospheric environment* 244 (2021): 117834.

Unterstrasser, Simon. "Properties of young contrails—a parametrisation based on large-eddy simulations." *Atmospheric Chemistry and Physics* 16.4 (2016): 2059-2082.

Reviewer #2 (Remarks to the Author):

1. Noteworthy Results

The manuscript presents the first large-scale, observation-based quantification of the net radiative forcing (RF) of contrails embedded within cirrus clouds, using a dataset of approximately 40,000 cases.

The study finds that embedded contrails have a predominantly warming effect, with a mean net RF of approximately 60 mW m^{-2} , and that this effect is especially pronounced at night.

The research demonstrates that the radiative impact of embedded contrails can be of similar magnitude to that of traditional line-shaped contrails and contrail cirrus, highlighting an important and previously underappreciated component of aviation's climate impact.

The analysis also captures the reduction in RF during the Covid-19 pandemic, providing a unique real-world test of the sensitivity of atmospheric RF to changes in air traffic.

2. Significance to the Field

This work is highly significant for the atmospheric science and climate modeling communities, as it fills a notable gap in the quantification of aviation-induced cloud effects.

The findings will be of interest not only to those studying aviation impacts, but also to researchers focused on cloud microphysics, remote sensing, and climate mitigation strategies.

Compared to the established literature (e.g., Burkhardt & Kärcher, 2011; Lee et al., 2021; Singh et al., 2024), this study provides a novel observational perspective and extends previous modeling and limited-case studies to a much larger and more representative dataset.

The work is original and does not duplicate existing studies.

3. Support for Conclusions

The conclusions are generally well supported by the data and analysis presented.

The use of both perturbed and unperturbed cloud regions for comparison is a strength, as it allows for a direct estimate of the incremental effect of embedded contrails.

The claims about the significance of embedded contrails to overall aviation RF are justified; however, the extrapolation from local to global effects is acknowledged as a limitation and could be discussed further.

We agree that we have been ambiguous regarding the meaning of the numbers presented. In line with the comments of Reviewer 1, who shared a similar concern, we have added the following discussion to Section 3 after explaining the differences between model- and observation-based estimates of RF (see Reviewer 3, first comment):

“While the inferred average net RF of 60 to 110 mW m^{-2} (depending on considered time period) is related to global observations, it refers to the local effect at the level of individual clouds. We follow two approaches for transforming those values into an estimate of the annual global mean net RF of embedded contrails. Both try to weigh the local net RF by assuming that it can only contribute to the global mean net RF where embedded contrails are present. For a first estimate, we re-grid the data in Figure 1a and b to 1° by 1° resolution (as 2.5° by 2.5° grid boxes overrepresent regions with low flight density) to obtain an embedded contrail fraction of around 0.10. We then scale the findings in Table 1 by this number and get values in the range from 5 to 11 mW m^{-2} . For a second estimate, we include empty grid boxes in Figure 1a and b in the averaging, assuming they

contribute a RF of zero to the global average. This gives a globally averaged net RF in the range from 4 to 8 mW m⁻². Knowledge of the distribution of embedded contrail cover would improve that admittedly crude estimate, but it provides a first observational constraint on a previously unquantified non-CO₂ effect of aviation. Overall, our results suggest that the climate effect of embedded contrails is on the order of 10% of the effect of line-shaped contrails (Lee et al, 2021) and, thus, a non-negligible contributor to the climate impact of aviation.”

Those estimates are also provided in the Abstract whose second part now reads:

“The annual mean local net RF of individual embedded contrails ranges between -320 mW m⁻² (2020, Covid lockdown) and 160 mW m⁻². Considering all cases from 2015 to 2021, we find an annual mean local warming effect of 60 mW m⁻². The expansion of our findings to the global scale suggests an annual global mean net RF of embedded contrails on the order of 5 mW m⁻². This corresponds to around 10% of the current estimate of the climate impact of line-shaped contrails and, together with recent findings that conditions for contrail formation are found most often in already-existing cirrus, suggests that embedded contrails are a non-negligible contributor to aviation’s impact on climate.”

4. Data Analysis, Interpretation, and Conclusions

The data analysis is thorough and uses appropriate statistical methods.

The interpretation of the results is logical and consistent with the data.

There are no major flaws that would prohibit publication, but the authors should clarify the limitations associated with the look-up table approach and the potential impact of one-dimensional radiative transfer assumptions.

The discussion of uncertainties is adequate, though a more explicit comparison with the uncertainties in previous studies would be beneficial.

We do agree with the Reviewer that the limitations of the LUT approach and three-dimensional radiative effects are important factors to consider when estimating the cirrus cloud RE. Section "Uncertainties in the estimate of net RF" partly discusses the limitations of the LUT approach, mainly the restricted parameter space and parameter spacing. The limited parameter space is explained and justified using aircraft observations by Krämer et al. (2016, 2020). Uncertainties in the CALIPSO retrieval products are also briefly discussed. We now point more explicitly towards Wolf et al. (2023a) for details regarding the sensitivities and uncertainties of the used LUT.

We have restructured and revised Section 5 for clarity. We have also added further details regarding three-dimensional effects to that section:

"Finally, systematic biases in cloud RE may also be caused by the assumption of plane-parallel clouds in the one-dimensional (1D) RT calculations used to create the LUTs. While this assumption may be suited for aged and stratiform contrails, it is violated in case of younger and line-shaped contrails. However, 1D and 3D RT calculations of cirrus and contrails are consistent over a wide range of conditions with largest differences for large SZA, i.e., during sunrise and sunset (Gounou and Hogan, 2007; Forster et al, 2012; Carles et al, 2025). The observations used in our study occur rarely, less than 5%, near sunrise and sunset."

5. Methodology

The methodology is sound and meets the standards expected in the field.

The use of spaceborne lidar, combined with aircraft position data and robust radiative transfer modeling, is appropriate and innovative.

The criteria for identifying embedded contrails and the temporal/spatial matching are clearly explained.

6. Reproducibility

Sufficient detail is provided in the methods section to allow for reproduction of the study.

The authors have made their data and code openly available, which further enhances reproducibility.

Minor improvements could be made by including more explicit instructions or flowcharts for the data processing steps.

We restructured Section 4.4, added minor improvements, and included more details on the data processing steps:

“In our case, IWC, ICER, and COT_1 are obtained as median values with respect to the perturbed and unperturbed regions of the observed cloud (see Section 4.3). SZA, T_c , T_{srf} , α_{srf} , and ice crystal shape are obtained for the intercept points between CALIPSO ground track and the flight paths and, for simplicity, are used for both the perturbed and unperturbed regions.”

and

“Based on T_c and ISS (see Section 4.1), ice crystals are classified ...”

7. Additional Comments and Suggestions

Consider citing the recent review by Singh et al. (2024, ACP 24, 9219–9300 <https://acp.copernicus.org/articles/24/9219/2024/>) to further contextualize the results within the broader field.

We added a reference to Singh et al. (2024) to the Introduction.

The manuscript is generally well written, but some sections could be streamlined for clarity.

We have revised the manuscript according to the comments of the three referees to improve clarity.

The figures are informative, but captions could be expanded for standalone comprehension.

We are sorry if captions were not detailed enough for standalone comprehension of the figures. More specific feedback would have helped us in expanding the text accordingly. Nevertheless, we have double-checked all captions and found them to properly identify the presented parameters. We leave the discussion of the figures' results to the main text, as should be.

The limitations and future work sections could be strengthened by discussing how the approach might be adapted for global, area-averaged RF estimates.

Thank you for this suggestion. We believe that this item has been covered by the revisions related to the first comments of Reviewer 1 and yourself.

Recommendation:

The manuscript represents a significant and original contribution to the field. With minor revisions to address the points above, I recommend it for publication.

Reviewer #3 (Remarks to the Author):

General comment:

This manuscript presents a timely and important study on the climate impact of embedded contrails—those forming within pre-existing cirrus clouds. By combining aircraft position data with satellite lidar observations, the authors compile a substantial dataset to estimate the previously underexplored net radiative forcing (RF) of these contrails using radiative transfer modeling. The analysis reveals that embedded contrails have an overwhelmingly warming effect, with younger contrails contributing more strongly to warming, while persistent ones tend to have a cooling impact. The study offers meaningful insights for aviation induced climate impact assessments, addresses a gap in current understanding, and is methodologically sound. The manuscript is well-written, and I recommend publication after minor revisions.

Major comment:

1. In Sect. 2, regarding the selection of unperturbed cirrus. The description of “lower values and a rather homogeneous structure are found further away from the aircraft’s trajectory” raises a question—could these regions represent older, sedimented contrails? If such cases are excluded from the statistics, and given that they tend to have lower or cooling RF as discussed later, their omission may lead to an overestimation of the overall RF.

Indeed, regions further away from an aircraft’s trajectory could have been affected by another, earlier aircraft. This holds particularly in regions of high air-traffic density. Ultimately, the question comes down to whether we are looking at (i) the total effect of embedded contrails (using a world without air traffic as reference) or (ii) a more incremental effect of individual aircraft. Modeling studies can be used for addressing the first point as they allow for contrasting simulations with and without contrails. Our observation-based method deals with the second point because we lack an aviation-free reference. In our concept, sedimented contrails in the reference region actually contribute to the cloud background when inspecting perturbances related to individual aircraft for their local RF but our method does not require identifying or quantifying that contribution. However, this has implications for the interpretation of the results. The contamination of the reference region with remnants of earlier contrails decreases the contrast between the perturbed and unperturbed signals compared to a well-defined single contrail in an unperturbed cloud. Based on the fact that contrail formation quenches in-cloud supersaturation one can assume that the overall perturbation of a cloud becomes less pronounced the more aircraft fly through it. Compared to the modeling framework with a clean reference, our observational data might therefore represent a lower estimate of RF.

This is an important aspect of our work. We have thus added the following statement to the discussion of the results to put our findings onto perspective:

“In contrast to modeling studies that compare simulations with and without contrails to assess their overall effect, our observation-based analysis focuses on the local impact of individual aircraft. In this scenario, the case-specific cloud background might actually contain perturbations from earlier aircraft. Those would reduce the contrast between perturbed and unperturbed cloud region compared to a lone-aircraft scenario. In reality, however, it is more likely that the impact of multiple aircraft flying through the same cloud scales non-linearly and eventually saturates so that later perturbations become indistinguishable from the background. This is also reflected in our data set when inspecting subsequent cases in regions of high air-traffic density (not shown). One might thus consider our estimate of net RF as a lower estimate in absolute terms.”

2. In Sect. 3, the statement that “the net RF is controlled primarily by ICER as this parameter affects both the solar and the thermal infrared components” could benefit from further clarification. If I

understand correctly—particularly in reference to the Nakajima and Nakajima (1999) bi-spectral method used to retrieve optical depth and effective radius—ICER mainly influences the thermal infrared component, while its impact on the solar component is more limited.

We agree with the Reviewer that the statement was not phrased precisely, and the text has been rephrased to avoid misunderstandings. We have rephrased the statement in Section 3 for clarification:

"Wolf et al (2023a) demonstrate that the cloud RE is most sensitive to changes in ICER and IWC and less so to changes in Sun position, surface albedo, and surface temperature. However, the cloud RE also depends on the absolute values of ICER and IWC and, therefore, their individual combinations. While this ambiguity complicates a potential parameterization of net RF to contrail-related changes in cloud microphysical properties, ICER and IWC are regarded as the main drivers of the solar and terrestrial components as well as cloud RE."

3. In Sect. 4.4, the estimation of the net RF of embedded contrails is based on look-up tables (LUTs). Could you clarify the assumptions or settings used for temperature, humidity, and overall atmospheric profiles in these LUTs?

Three atmospheric profiles, including temperature and relative humidity, were used in the simulations to represent the US-standard atmosphere (afglus), a tropical atmosphere (afglt), and a sub-arctic winter atmosphere (afglsw), spanning from warm to cold climates. These atmosphere profiles were taken from *Anderson et al. (1986)*. All details about the simulation setup, the simulated parameter space, and related uncertainties are given in *Wolf et al. (2023a)*. To keep the manuscript concise, we refrain from adding too many details about the creation of the LUT here.

The beginning of Section 4.4 has been modified as follows:

"The estimation of the net RF of embedded contrails is based on the look-up tables (LUTs) of Wolf et al. (2023a,b). Details of and simplifications in those calculations can be found therein. Differences related to meteorological conditions used in the LUTs and for the observations cancel out in first approximation as we focus on the difference between the radiative calculations with and without embedded contrails."

Minor comment:

1. In Sect. 2, when introducing the net RF, the phrase “The net RF of the embedded contrail (ΔF_{net})” should include a link to the Methods section, where the calculation or definition is provided.

We followed your recommendation and included a link to Section 4.4, where the calculation or definition of ΔF_{net} is provided. The sentence now reads:

“The local net RF of the embedded contrail (ΔF_{net} ; see Section 4.4 for details) contains...”

2. In Sect. 2, the manuscript states that global matches between aircraft waypoints and CALIOP observations for 2012 are selected as the first period, while the second period focuses on flights to and from Europe from 2015 to 2021. It would be helpful to clarify in the main text why 2012 was chosen as the reference year and why this period covers global data, whereas subsequent years are geographically limited to Europe. Although these details are partly provided in the Methods, a brief explanation in the main text would avoid the abrupt impression.

The two periods and regions covered in our study are linked to the availability of aircraft waypoint data sets, which are an essential input to the analysis. The one for 2012 is similar to those used in, e.g., *Bedka et al. (2013)* and *Duda et al. (2019)* and contains global data. The one for 2015 to 2021 is provided by EUROCONTROL and contains only flights from and to Europe during four months

of each year. While this information is provided in the methods section, we agree that the seemingly random selection of regions and years requires clarification in the main text as well. To be more specific, we have thus changed the statement in question to:

“Matches of aircraft locations and CALIOP observations are investigated for two time periods that are constrained by the availability of waypoint data for individual aircraft (see Sections 4.1 and 4.2 for details).”

To further highlight the usefulness of investigating the shared region, we have added the following statement to the end of the same paragraph:

“Incidentally, this shared region includes the North Atlantic Corridor and Europe, which were recently found to be areas in which conditions for cirrus occurrence almost always overlap with conditions for forming long-lived contrails (Petzold et al., 2025).”

3. In Sect. 3, “besides a seasonal variation, an increasing trend in net RF from negative to positive is found for the time period from 2012 to 2019”. Changing to “2012 and 2015 to 2019” would be more correct.

We agree with the reviewer and have corrected the sentence to:

“Besides a seasonal variation, an increasing trend in net RF from negative to positive is found for the time period from 2012 and 2015 to 2019.”

References cited in the response to all reviewers:

Anderson, G. P., Clough, S. A., Kneizys, F. X., Chetwynd, J. H., and Shettle, E. P. (1986), AFGL atmospheric constituent profiles, Environ. Res. Pap., 954, p. 1-46.

Bedka, S. T., Minnis, P., Duda, D. P., Chee, T. L., and Palikonda, R. (2013), Properties of linear contrails in the Northern Hemisphere derived from 2006 Aqua MODIS observations, Geophys. Res. Lett., 40, <https://doi.org/10.1029/2012GL054363>.

Dobbie, S. and Jonas, P. (2001), Radiative influences on the structure and lifetime of cirrus clouds, Quart. J. Royal Meteorol. Soc., 127, <https://doi.org/10.1002/qj.49712757808>.

Duda, D. P., Bedka, S. T., Minnis, P., Spangenberg, D., Khlopenkov, K., Chee, T., and Smith Jr., W. (2019), Northern Hemisphere contrail properties derived from Terra and Aqua MODIS data for 2006 and 2012, Atmos. Chem. Phys., 19, <https://doi.org/10.5194/acp-19-5313-2019>, 2019.

Forster, L., Emde, C., Mayer, B., and Unterstrasser, S. (2012), Effects of three-dimensional photon transport on the radiative forcing of realistic contrails, J. Atmos. Sci., 69, 2243-2255, <https://doi.org/10.1175/JAS-D-11-0206.1>.

Gounou, A. and Hogan, R. J. (2007), A sensitivity study of the effect of horizontal photon transport on the radiative forcing of contrails, J. Atmos. Sci., 64, 1706-1716, <https://doi.org/10.1175/JAS3915.1>.

Krämer, M., Rolf, C., Luebke, A., Afchine, A., Spelten, N., Costa, A., Meyer, J., Zöger, M., Smith, J., Herman, R. L., Buchholz, B., Ebert, V., Baumgardner, D., Borrmann, S., Klingebiel, M., and Avallone, L. (2016), A microphysics guide to cirrus clouds – Part 1: Cirrus types, Atmos. Chem. Phys., 16(5), 3463-3483, <https://doi.org/10.5194/acp-16-3463-2016>

Krämer, M., Rolf, C., Spelten, N., Afchine, A., Fahey, D., Jensen, E., Khaykin, S., Kuhn, T., Lawson, P., Lykov, A., Pan, L. L., Riese, M., Rollins, A., Stroh, F., Thornberry, T., Wolf, V., Woods,

S., Spichtinger, P., Quaas, J., and Sourdeval, O. (2020), A microphysics guide to cirrus – Part 2: Climatologies of clouds and humidity from observations, *Atmos. Chem. Phys.*, 20, 12569-12608, <https://doi.org/10.5194/acp-20-12569-2020>

Petzold, A., Khan, N. F., Li, Y., Spichtinger, P., Rohs, S., Crewell, S., Wahner, A., and Krämer, M. (2025), Contrails inside cirrus clouds predominate with uncertain climate impact. *Research Square* [Preprint](version 1). <https://doi.org/10.21203/rs.3.rs-6837438/v1>

Teoh, R., Engberg, Z., Schumann, U., Voigt, C., Shapiro, M., Rohs, S., and Stettler, M. E. J. (2024b), Global aviation contrail climate effects from 2019 to 2021, *Atmos. Chem. Phys.*, 24, <https://doi.org/10.5194/acp-24-6071-2024>.

Tesche, M., Achtert, P., Glantz, P., and Noone, K. (2016), Aviation effects on already-existing cirrus clouds, *Nature Comms.*, 7, <https://doi.org/10.1038/ncomms12016>.

Wolf, K., Bellouin, N., and Boucher, O. (2023a), Sensitivity of cirrus and contrail radiative effect on cloud microphysical and environmental parameters, *Atmos. Chem. Phys.*, 23, <https://doi.org/10.5194/acp-23-14003-2023>.

Wolf, K., Bellouin, N., and Boucher, O. (2023b), Simulated top-of-atmosphere (120 km) downward and upward solar and thermal-infrared irradiances and ice cloud optical thickness; calculated solar, TIR and net cloud radiative effect. Simulated with ice crystal properties for aggregates, droxtals, and plates based on Yang (2013), <https://doi.org/10.5281/zenodo.8159286>.

We again thank the reviewers for their constructive suggestions for improving the clarity of the paper. Please find our detailed replies (blue text) and changes to the text (blue italic) below.

Before providing our replies to the Reviewers' comments, we would like to address an issue that came up since the last submission and has been addressed during the revision of the manuscript. We have identified a programming mistake in the code for calculating the globally averaged net RF in the second approach. After solving the problem, we have revised the annual global mean net RF in the text accordingly. The text now reads:

“For a second estimate, we also re-grid the data in Figure 1a and b to a 1° by 1° resolution and include empty grid boxes in the averaging, assuming they contribute a RF of zero to the global average. This gives a globally averaged net RF in the range from 6 to 9 mW m⁻².”

Reviewer #1 (Remarks to the Author):

Since this is a second review, I have restricted my comments to concerns which were raised during the first round.

I would like to thank the authors for their careful and comprehensive response to my previous comments. The updated abstract is dramatically improved, and my concerns regarding local vs global warming have been fully addressed. I also consider their response regarding the methodological questions adequate.

However, I would request one additional change. I was a little confused by the response regarding longer-term changes (i.e. those which do not manifest within 30 minutes of an aircraft's passage), which discusses only the possibility of changes in humidity/lapse rate and cirrus cloud adjustments and states that "disentangling these processes would require a model capable of representing embedded contrail and cirrus evolution". However, the later author response stated that the time delay was kept below 30 minutes because "this is the time period for which even a strongly advected contrail should still be within the perturbed section of the CALIPSO track". These responses seem inconsistent, and knowing why this 30 minute limit exists could be important to the reader. Specifically, I would be grateful if the authors could clarify the degree to which the 30 minute limit is dominated by wind speed error (i.e. the inability to accurately calculate flight track advection), the width of the CALIPSO track, and the lack of an accurate model of contrail/cirrus evolution. This could help to prioritize research, given that each of these issues is resolvable but with very different approaches. Since this is mostly a clarification I would consider this a minor change, but one I hope that the authors will consider as it would help increase both the readability and direct utility of the manuscript.

Thank you for giving us the opportunity to elaborate on this issue. The most critical point of our approach is to know the precise location of where to look for an embedded contrail within the CALIPSO measurement. During our initial work (Tesche et al., 2016) we concluded that 30 minutes delay ensures that the contrail signature should still be within the part of the measurement that we consider as perturbed. For longer time periods, we cannot rule out that the processes mentioned in our reply dilute the contrail signature. Such scenarios would require additional information for constraining the analysis. Adding complex model simulations to every potential case not only increases the computational effort but also complicates the interpretation of potential findings by adding further degrees of freedom. As a potential next step for extending the time delay beyond 30 minutes, we consider using forward trajectories from the position of the aircraft to better constrain the location of the contrail in the CALIPSO curtain.

We have included the last sentence in the section “Discussion”:

“As a potential next step for extending the time delay beyond 30 minutes, we consider using forward trajectories from the position of the aircraft to better constrain the location of the contrail in the CALIPSO curtain.”

Reviewer #3 (Remarks to the Author):

I appreciate the authors' efforts in improving the careful interpretation of the annual mean net RF of embedded contrails.

I have only one remaining question regarding the use of an embedded contrail fraction of around 0.10. This appears to be a significant scaling factor for the global net RF estimate. The authors state:

“For a first estimate, we re-grid the data in Figure 1a and b to 1° by 1° resolution (as 2.5° by 2.5° grid boxes overrepresent regions with low flight density) to obtain an embedded contrail fraction of around 0.10.”

Are there any references or justifications supporting the choice of a 1° by 1° resolution for this regridding step?

With this minor revision, I would support the manuscript's acceptance.

Thank you for the positive feedback. The reviewer is correct that grid resolution has a crucial impact on the forcing estimate. In fact, it increases almost linearly with decreasing (coarser) grid resolution. We have chosen a resolution of 1° by 1° based on the resolution of the CALIPSO measurements. In the analysis, we consider 15 lidar profiles around the aircraft-CALIPSO intercept. Each profile has a 5 km along-track resolution, resulting in a total of 75 km, referred to as the observation length scale (OLS). Using 1° by 1° resolution, grid boxes are then about 110 km by 110 km at the equator and OLS by 110 km at the mid-latitudes. A finer resolution, would underrepresent the OLS. In contrast, 2.5° by 2.5° resolution represents grid boxes of about 280 km by 280 km at the equator and 2 times the OLS by 280 km at the mid-latitudes. In addition, those cells might extend well into areas with very low occurrence rates of embedded contrails.